# Peer review of "Bio-inspired design of ice-retardant devices based on benthic marine invertebrates: the effect of surface texture"

_PeerJ, doi:10.7717/peerj.588_

## Round 0.1 · original submission · Major Revisions

please address the comments and incorporate the suggestions from two reviewers in the revision.

Reviewer 1 ·

Basic reporting

Seems fine.

Experimental design

Seems fine.

Validity of the findings

Seems fine.

Additional comments

This manuscript deals with the formation of ice crystals on submerged and emerged biogenic structures such as shells and calcareous spines. The authors have created a series of 3D printed plastic generic surfaces that replicate surface features of three marine bivalves and a seastar. The approach of using a standardized material to remove the effects of surface coatings is a sensible one, and the authors use it to illustrate the relatively minor effects that these small scale changes in rugosity and surface pattern contribute to delaying ice formation. The use of the 3D printing also allows the authors to vary the spacing and height of the surface features to combinations that are not necessarily found on present-day members of these species, making it possible to explore more of the parameter space than would be available with only natural samples. The authors conclude that the small changes in texture used in the trials add up to small changes in freeze time for seawater on the surfaces, both submerged and in air.
On the whole I think the manuscript is well written, concise, and clear. There are only a few small points I would like to see addressed to help clarify some aspects of the experimental procedure.
In figure 1, the detail of the four samples (b, d, f, h) are difficult to make out, even when zoomed in. I think it’s necessary to redraw those four pieces at a larger size in order to better show the surface texture. In addition, the text and figure don’t make it terribly clear how the valley and stripe textures differ, beyond being rotated 90 degrees from one another in the figure. I’m guessing that the valley pattern is more like a triangle wave, while the stripe pattern is more akin to a square wave, but it would be helpful to more explicitly describe how these two pieces differ.
In figure 2, the sudden appearance of red coloration is unexplained. Was all of the artificial seawater also dyed? Or were these drops only colored for the purposes of illustration in the figure? Please clarify this, as readers may want to know if other things were being added to the seawater besides the usual salt mixture.
In figures 3 + 5, please state what the error bars represent.

Finally, I would like to commend the authors on using a 3D printer for something worthwhile. If the internet is to be believed, 3D printers are only used to produce comic book figurines and things to stick in someone’s stretched earlobes.

·

Basic reporting

For all figures:
For all data figures, I would much prefer to see 95% confidence intervals or 2 x standard error and not standard deviation.
For all figures, due to the low sample size and high variability, I would prefer to see actual data points instead of the mean or a summary statistic.
In general, the figures and captions do not stand alone. Although they need not describe absolutely everything about the experiment, slightly more information would be helpful. Specifically relating figures back to experiments by using the established experiment name would be helpful.
Table 1
Must present Ns.
Figure 1
Scaling looks awry for Pisaster. Please double check that it isn’t.
Figure 2
(b)(c) At no point do the authors indicate what makes the droplets of seawater red, or if the coloring agent itself has any effect on the growth of ice.
Figure 3
The findings presented in Figure 3 are potentially the most substantial and interesting findings in the paper – at least from a biological perspective. Because of this, the authors must exercise extreme care in reporting and interpreting these data. However, this figure, and perhaps the analysis of the data and the conclusions derived therewith, appear to be flawed. Further, the data collected in this paper do not appear to have a negative control (seawater). At minimum, the figure and conclusions derived from it require substantial revision or should be omitted from the paper.
Primarily, the direct comparison of the data from Denny and from the present study appears to be flawed and probably shouldn’t be done. For one, Denny appears to have mislabeled both species and bars, and the caption and the text of the results conflict (Sterechinus is not a sponge and Suberites is not an urchin). If this paper chooses to incude that data, it should be corrected or a note should be added. Secondly, in the original source figure 3, Denny reports “trials with nucleation occurring” on the y-axis, which is presumably out of 6 -9 trials. In the present paper, the Denny data appears to be incorrectly transformed to “proportion forming ice” on the y-axis, with Suberites now reaching to 100%. This transformation would not be correct given 5 trials out of 6 – 9 total trials, and further, the text of the caption supports that the value for Suberites is not 100% “[it] formed anchor ice in [only] nearly every trial”. It is clear that the figure from Denny is unclear, and because of this, it should perhaps not be meddled with or even included in this paper.
Further, in the caption, the criteria (time until ice formation?) for marking positive or negative should be emphasized. Were the same criteria used the same for both previous and current data? This is not clear to me from reading both papers.
Regarding only the data collected in the present paper and presented in Figure 3, the results cannot be assessed independent of a negative control (seawater), which does not appear to have been conducted. Without a negative control of the bathing seawater that never nucleates, it is impossible to say whether something in the artificial seawater or beaker or cold room is causing premature nucleation. This experiment absolutely must be controlled if it is to be reported
Another note for figure 3: If the Denny data is correctly presented and included in this paper, the caption (and the methods/discussion), differences between the two experimental protocols should be explicitly stated (that one started with samples at 4˚C and one at 1˚C, that one used artificial seawater and one natural Antarctic seawater from McMurdo Sound).
In the caption and in the text of the paper, it should be highlighted that beakers for both experiments were uncovered. In my experience, cold rooms at -20˚C are replete with suspended ice crystals in air currents and this is the likely source of nucleation of these beakers – not the organisms themselves.
Because the water and alcyonium bars are both 0 and have no color, the presented color scheme doesn’t work.
Whether bars are means and indicate SD or another summary statistic must be indicated.
Figure 4.
It would be preferable to present 2xSE or 95% confidence interval as opposed to SD. N must be presented. With the small dataset and large variation, it would also be better to show each and every data point for each texture as opposed to a summary value (mean). It would also be good and interesting to have an untextured plate as a control, but this is not required.
Figure 5.
It is imperative that the mean time of the control, to which all data is normalized, be stated in the caption and perahaps in the text. It would be preferable to show 95% CI or 2SE and not SD. Whichever is used, it must be stated in the caption (it is not). N must be presented.

Experimental design

It is unclear if the experiments are appropriate. See comments in "Validity of the findings" below.

Validity of the findings

It is unclear whether the experiments presented herein were conducted rigorously and to a high technical standard. I have several general concerns, primarily relating to the methods. These concerns revolve primarily and importantly around the high likelihood that a -20˚C cold room will contain innumerable ice nuclei wafting around in air currents. Since none of the experiments appear to have been covered or otherwise protected from airborne ice nuclei (aside from siting away from overhead fans) it is impossible to tell if the source of the ice formation has any relation to the surface being tested or if it is simply a stochastic process resulting from chance encounters of preexisting ice crystals in the air with undercooled water in the experiments. To rule this out, I would strongly suggest repeating or supplementing the experiments and including transparent covers to protect against exogenous ice inoculation that do not have to be removed in order to observe ice formation. If the results hold, then they are likely valid and that is interesting. If not, they do not.

My most important criticism is relates to the findings from the ice formation on sub-polar organisms, since these results are potentially the most significant, if true. This experiment absolutely must be controlled for using the same artificial seawater that was used in the experiments, otherwise the data should absolutely not be presented in this paper. Furthermore, as per my comments about the figure above, great care must be taken when comparing data from this study to those from Denny. At present, it does not appear that the comparisons were correctly made. Finally, the differences in the experimental protocol between Denny and this experiment (namely the starting temperature of the water and type of seawater) must be emphasized since they are different – so a direct comparison cannot be made. If a comparison to the Denny paper must be made (and the potentially flawed and certainly confusing Denny data is to be included in this paper) it would be best if the experiments were repeated with the exact same conditions (excluding the Antarctic Seawater, of course). Repeating the experiments is not absolutely necessary if the discrepancies in the experimental design are highlighted, although performing a control experiment is. If the above concerns cannot be assuaged then experiments relating to the organisms should be removed from the paper or, at the very least, the statements of findings should be extensively qualified and these issues discussed in the paper.

Additional comments

I see value in ultimately publishing the results from this study. However, in its current form I find it difficult to interpret the findings. Specifically, I find that some of the major results of the paper are not adequately controlled, some of the experiments may be flawed, and the conclusions, as presented, are therefore not adequately qualified. Given some of my above stated criticisms of experimental design, the authors should either repeat some of the experiments or consider and discuss in the manuscript the alternate reasons why they might have obtained the results that they did. I can conceive of scenarios in which adequate controls and modified methods could in fact enhance the magnitude of the differences seen between differentially textured plates, or lessen them. Even if experiments reported herein cannot be repeated in order to address my concerns - for e.g., because of funding or personnel issues - it is imperative that the limitations to the techniques used herein be adequately discussed so that future studies can continue to improve on the methodology.

I have spent much time thinking about these issues and attempting experiments of this sort over the years. Please fee free to contact me if you would like to discuss my concerns with the paper in person as you work to revise and resubmit.

Specific comments (not exhaustive):
L 11: Ecomechanical is not a part of the Biology lexicon, or at least is not well known. Define or substitute.
L 12: What did they find about the plants? Why include this?
L 30: Cillia covering surfaces of echinoderms and cnidarians should perhaps be included in this discussion.
L 18: Subcooled is not correct. It should be “undercooled” or “supercooled”.
L 35. New paragraph before “We hypothesized…”
L40: It would strongly contest that Pisaster is hard-shelled. It is composed of bony plates covered with an epithelium – perhaps like your elbow or skull. That is has an epithelium covering its entire surface (most likely with cilia and mucus throughout) makes it altogether different from the bivalves tested.
L55. The differences between the tests in the two studies should be highlighted. They are not the same.
L75. Typo “setting designing”
L110: Were the bilvalve shells open or closed? Did the ice grow from the inside or on the shell on the exterior?
L135: It is not clear how glacier scour relates to this paper or its findings? It probably doesn’t.
L136: “Unfortunately….”. It is not clear how this statement is relevant.
L150. How do feathers, presumably on endothermic birds, have anything to do with this study?
L155: The echinoderms have calcareous ossicles, but they are covered completely with a presumably ciliated epithelium. This makes them entirely different than the shells of the bivalves, and perhaps more like the living tissue inside the bivalves.
L169. It is not clear how shark skin is relevant to this paper.

---

## Round 0.2 · Minor Revisions

The reviewer found some typos and made some suggestions. Please incorporated these comments in the revised manuscript.

·

Basic reporting

Ok. See General Comments.

Experimental design

Ok, subject to the limitations discussed in the text. See General Comments.

Validity of the findings

Ok, subject to the limitations discussed in the text. See General Comments.

Additional comments

General Comments:
The authors have clarified or revised the major issues from the initial draft of the manuscript. However, a number of small, but important, issues remain that should be addressed prior to publication. Although there are some inherent limitations to the methodology, the limitations are discussed and the results are adequately qualified in the text.

Despite the limitations, the paper provides an important stepping-stone towards further understanding of the determinants of ice formation on surfaces and marine animals, and should be included in the literature.

Specific comments

In the abstract, it is not clear what organisms were screened, nor the results of those studies. Consider rewording one sentence to:
“…we screened sub-polar marine organisms and artificial 3-D printed samples…”

However, note that with the biological tests the authors are not testing surface texture alone, so might need more extensive rewording.

In the abstract, the results of the biological tests are not described. Either remove these results from the paper or include mention of the results in the abstract.

In the abstract, a tiny note on the limitations of these tests, as described in the discussion [and further below here] would be appropriate.

The introduction appears to be missing the rationale for testing biological samples, and whether or how that relates to the overarching hypothesis that surface texture alone is responsible for differing susceptibility. (For the 3 hard shelled mollusks tested in this study, perhaps this is indeed a test of surface texture alone, but the echinoderm confounds that line of reasoning).

L45 - Move “and” after specific heat.

L46 – “an” typo

L55 – The authors should lump the three molluscs into a category and describe them as bivalve molluscs to differentiate them from the echinoderm, which should come separately and be labeled as such.

L69 – “…submerged surface of the sample [add something about the length of the assay]…” – they will all grow ice eventually at -20˚C).

L70 – Are they using the same time criteria as Denny?

L70 and following - Could starting temperature have any effect on that, perhaps by a warmer starting sample becom ing more thermally stratified in a -20˚C cold room and thus cool faster over all due to enhanced convective mixing?

L71 – “to a” typo

Results – Results for biological samples have not been described in the abstract.

L128 – Within what time?

L125. Because the Denny paper and this paper contradict each other in reporting the data presented in figure 3, the authors must state that the Denny data in figure 3 was compiled by reanalyzing primary data (as told to me by the authors), and thus is more correct than the same data presented in the original figure in the original source; where the two figures do not agree, this one is more correct.


Discussion:
Limitations of the ice formation test: This is an appropriate and important addition. Thank you for adding it. Due to its extensive nature, a tiny mention of limitations would be appropriate in the abstract, as in something like, “despite limitations inherent to our techniques, surface texture alone was observed to play a small role in resistance to ice formation…”.

Discussion, Sub-polar species always initiated ice formation:
The authors should highlight that 3 out of the four species that they tested are distinctly different than any of the species tested in the previous paper by Denny. Denny didn’t test any molluscs nor any animals with dead exoskeletons, while 75% of the species tested in this study are bivalve molluscs with dead, inanimate calcium carbonate shells. The authors might balance this by highlighting the difference observed between the - at least superficially similar - Odontaster (Antarctic sea star) and Pisaster (Sub-polar sea star).

Discussion: The authors should consider whether the difference in ice formation observed for the two sea star species between the two studies may also be due to cold-induced inactivation of cilia or other behavior for a sub-polar species held at -2˚C or colder.

Figure 2 caption: Identify the red so the caption stands alone. “…shift from red (middle arrow; contrast agent red food coloring added) to opaque white”, or some variation of this.

Figure 3. Caption: Time? Note Denny data is from reanalyzing primary data, not reproducing the figure. Control should say McMurdo Seawater (or McMurdo SW for short).

Figure 5 caption: Consider rewording to “Droplet test, freeze time normalized to mean of flat plate controls (834 s) vs. …” Then remove the same info from the last sentence.

---

## Round 0.3 · accepted · Accept

your manuscript is accepted.